# Diffusion Probabilistic Modeling for Video Generation

## Abstract

Denoising diffusion probabilistic models are a promising new class of generative models that mark a milestone in high-quality image generation. This paper showcases their ability to sequentially generate video, surpassing prior methods in perceptual and probabilistic forecasting metrics. We propose an autoregressive, end-to-end optimized video diffusion model inspired by recent advances in neural video compression. The model successively generates future frames by correcting a deterministic next-frame prediction using a stochastic residual generated by an inverse diffusion process. We compare this approach against six baselines on four datasets involving natural and simulation-based videos. We find significant improvements in terms of perceptual quality and probabilistic frame forecasting ability for all datasets.

## 1 Introduction

The ability to anticipate future frames of a video is intuitive for humans but challenging for a computer (Oprea et al., 2020). Applications of such *video prediction* tasks include anticipating events (Vondrick et al., 2016a), model-based reinforcement learning (Ha & Schmidhuber, 2018), video interpolation (Liu et al., 2017), predicting pedestrians in traffic (Bhattacharyya et al., 2018), precipitation nowcasting (Ravuri et al., 2021), neural video compression (Han et al., 2019; Lu et al., 2019; Agustsson et al., 2020; Yang et al., 2020; 2021a; 2022), and many more.

The goals and challenges of video prediction include (i) generating multi-modal, stochastic predictions that (ii) accurately reflect the high-dimensional dynamics of the data long-term while (iii) identifying architectures that scale to high-resolution content without blurry artifacts. These goals are complicated by occlusions, lighting conditions, and dynamics on different temporal scales. Broadly speaking, models relying on sequential variational autoencoders (Babaeizadeh et al., 2018; Denton & Fergus, 2018; Castrejon et al., 2019) tend to be stronger in goals (i) and (ii), while sequential extensions of generative adversarial networks (Aigner & Körner, 2018; Kwon & Park, 2019; Lee et al., 2018) tend to perform better in goal (iii). A probabilistic method that succeeds in all the three desiderata on high-resolution video content is yet to be found.

Recently, diffusion probabilistic models have achieved considerable progress in image generation, with perceptual qualities comparable to GANs while avoiding the optimization challenges of adversarial training (Sohl-Dickstein et al., 2015; Song & Ermon, 2019; Ho et al., 2020; Song et al., 2021c;b). In this paper, we extend diffusion probabilistic models for stochastic video generation. Our ideas are inspired by the principles of predictive coding (Rao & Ballard, 1999; Marino, 2021) and neural compression algorithms (Yang et al., 2021b) and draw on the intuition that residual errors are easier to model than dense observations (Marino et al., 2021). Our architecture relies on two prediction steps: first, we employ a deterministic convolutional RNN to deterministically predict the next frame conditioned on a sequence of frames. Second, we correct this prediction by an additive residual generated by a denoising diffusion process, also conditioned on a temporal context (see Figs. 1a and 1b). This approach is scalable to high-resolution video, stochastic, and relies on likelihood-based principles. Our ablation studies strongly suggest that predicting video frame residuals instead of naively predicting the next frames improves generative performance. By investigating our architecture on various datasets and comparing it against multiple baselines, we prove superior results on both probabilistic (CRPS) and perceptual (LPIPS, FID) metrics. In more detail, our achievements are as follows:

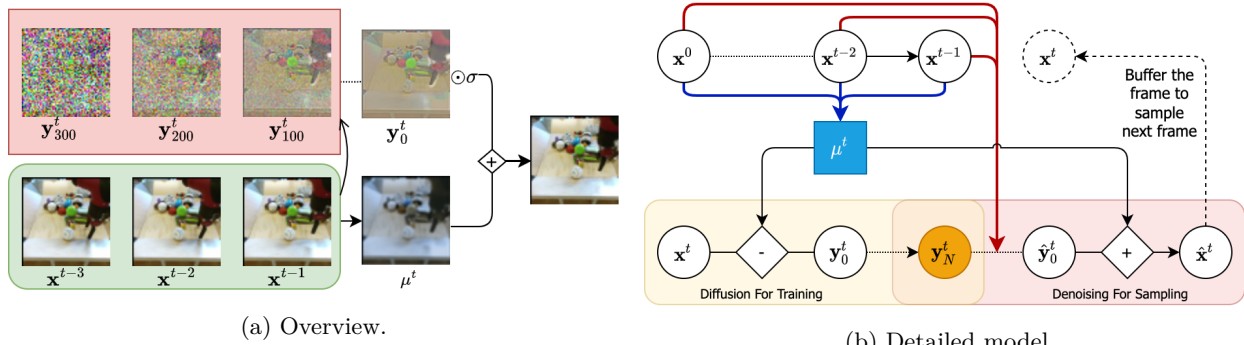

(a) Overview.  (b) Detailed model.

Figure 1: Overview: Our approach predicts the next frame $\mu_t$ of a video autoregressively along with an additive correction $\mathbf{y}_0^t$ generated by a denoising process. Detailed model: Two convolutional RNNs (blue and red arrows) operate on a frame sequence $\mathbf{x}^{0:t-1}$ to predict the most likely next frame $\mu^t$ (blue box) and a context vector for a denoising diffusion model. The diffusion model is trained to model the scaled residual $\mathbf{y}_0^t = (\mathbf{x}^t - \mu^t)/\sigma$ conditioned on the temporal context. At generation time, the generated residual is added to the next-frame estimate $\mu^t$ to generate the next frame as $\mathbf{x}^t = \mu^t + \sigma \mathbf{y}_0^t$.

1. We show how to use diffusion probabilistic models to autoregressively generate videos. This enables a new path towards probabilistic video forecasting while achieving perceptual qualities better than or comparable with likelihood-free methods such as GANs.

2. We also study our model based on three metrics, FVD/LPIPS/CRPS, that can cover both forecasting ability and perceptual quality. The result demonstrates that our model perform better than modern GAN and VAE baselines such as IVRNN, SVG-LP, SLAMP, RetroGAN, DVD-GAN and FutureGAN (Castrejon et al., 2019; Denton & Fergus, 2018; Akan et al., 2021; Kwon & Park, 2019; Clark et al., 2019; Aigner & Körner, 2018).

3. Our ablation studies demonstrate that modeling residuals from the predicted next frame yields better results than directly modeling the next frames. This observation is consistent with recent findings in neural video compression. Figure 1a summarizes the main idea of our approach (Figure 1b has more details).

The structure of our paper is as follows. We first describe our method, which is followed by a discussion about our experimental findings along with ablation studies. We then discuss connections to the literature, summarize our contributions.

## 2 A Diffusion Probabilistic Model for Video

We begin by reviewing the relevant background on diffusion probabilistic models. We then discuss our design choices for extending these models to sequential models for video.

### 2.1 Background on Diffusion Probabilistic Models

Denoising diffusion probabilistic models (DDPMs) are a recent class of generative models with promising properties (Sohl-Dickstein et al., 2015; Ho et al., 2020). Unlike GANs, these models rely on the maximum likelihood training paradigm (and are thus stable to train) while producing samples of comparable perceptual quality as GANs (Brock et al., 2019).

Similar to hierarchical variational autoencoders (VAEs) (Kingma & Welling, 2013), DDPMs are deep latent variable models that model data $\mathbf{x}_0$ in terms of an underlying sequence of latent variables $\mathbf{x}_{1:N}$ such that $p_\theta(\mathbf{x}_0) = \int p_\theta(\mathbf{x}_{0:N})d\mathbf{x}_{1:N}$. The main idea is to impose a diffusion process on the data that incrementally destroys the structure. The diffusion process's incremental *posterior* yields a stochastic denoising process

that can be used to *generate* structure (Sohl-Dickstein et al., 2015; Ho et al., 2020). The *forward*, or *diffusion* process is given by

$$q(\mathbf{x}_{1:N}|\mathbf{x}_0) = \prod_{n=1}^{N} q(\mathbf{x}_n|\mathbf{x}_{n-1});$$

$$q(\mathbf{x}_n|\mathbf{x}_{n-1}) = \mathcal{N}(\mathbf{x}_n|\sqrt{1-\beta_n}\mathbf{x}_{n-1}, \beta_n\mathbf{I}). \tag{1}$$

Besides a predefined incremental variance schedule with $\beta_n \in (0,1)$ for $n \in \{1,\cdots,N\}$, this process is parameter-free (Song & Ermon, 2019; Ho et al., 2020). The reverse process is called *denoising process*,

$$p_\theta(\mathbf{x}_{0:N}) = p(\mathbf{x}_N) \prod_{n=1}^{N} p_\theta(\mathbf{x}_{n-1}|\mathbf{x}_n);$$

$$p_\theta(\mathbf{x}_{n-1}|\mathbf{x}_n) = \mathcal{N}(\mathbf{x}_{n-1}|M_\theta(\mathbf{x}_n, n), \gamma\mathbf{I}). \tag{2}$$

The reverse process can be thought of as approximating the posterior of the diffusion process. Typically, one fixes the covariance matrix (with hyperparameter $\gamma$) and only learns the posterior mean function $M_\theta(\mathbf{x}_n, n)$. The prior $p(\mathbf{x}_N) = \mathcal{N}(\mathbf{0}, \mathbf{I})$ is typically fixed. The parameter $\theta$ can be optimized by maximizing a variational lower bound on the log-likelihood, $L_{\text{variational}} = \mathbb{E}_q[-\log\frac{p_\theta(\mathbf{x}_{0:N})}{q(\mathbf{x}_{1:N}|\mathbf{x}_0)}]$. This bound can be efficiently estimated by stochastic gradients by subsampling time steps $n$ at random since the marginal distributions $q(\mathbf{x}_n|\mathbf{x}_0)$ can be computed in closed form (Ho et al., 2020).

In this paper, we use a simplified loss due Ho et al. (2020) who showed that the variational bound could be simplified to the following denoising *score matching loss*,

$$L(\theta) = \mathbb{E}_{\mathbf{x}_0, n, \epsilon} ||\epsilon - f_\theta(\mathbf{x}_n, n)||^2$$

$$\text{where } \mathbf{x}_n = \sqrt{\bar{\alpha}_n}\mathbf{x}_0 + \sqrt{1-\bar{\alpha}_n}\epsilon. \tag{3}$$

We thereby define $\bar{\alpha}_n = \prod_{i=1}^{n}(1-\beta_i)$. The intuitive explanation of this loss is that $f_\theta$ tries to predict the noise $\epsilon \sim \mathcal{N}(\mathbf{0}, \mathbf{I})$ at the denoising step $n$ (Ho et al., 2020). Once the model is trained, it can be used to generate data by ancestral sampling, starting with a draw from the prior $p(\mathbf{x}_N)$ and successively generating more and more structure through an annealed Langevin dynamics procedure (Song & Ermon, 2019; Song et al., 2021c).

## 2.2 Residual Video Diffusion Model

Experience shows that it is often simpler to model *differences* from our predictions than the predictions themselves. For example, masked autoregressive flows (Papamakarios et al., 2017) transform random noise into an additive prediction error residual and boosting algorithms train a sequence of models to predict the error residuals of earlier models (Schapire, 1999). Residual errors also play an important role in modern theories of the brain. For example, predictive coding (Rao & Ballard, 1999) postulates that neural circuits estimate probabilistic models of other neural activity, iteratively exchanging information about error residuals. This theory has interesting connections to VAEs (Marino, 2021; Marino et al., 2021) and neural video compression (Agustsson et al., 2020; Yang et al., 2021a), where one also compresses the residuals to the most likely next-frame predictions.

This work uses a diffusion model to generate *residual corrections* to a deterministically predicted next frame, adding stochasticity to the video generation task. Both the deterministic prediction as well as the denoising process are conditioned on a long-range context provided by a convolutional RNN. We call our approach "Residual Video Diffusion" (RVD). Details will be explained next.

**Notation.** We consider a frame sequence $\mathbf{x}^{0:T}$ and a set of latent variables $\mathbf{y}^{1:T} \equiv \mathbf{y}_{0:N}^{1:T}$ specified by a diffusion process over the lower indices. We refer to $\mathbf{y}_0^{1:T}$ as the (scaled) frame residuals.

**Generative Process.** We consider a joint distribution over $\mathbf{x}^{0:T}$ and $\mathbf{y}^{1:T}$ of the following form:

$$p(\mathbf{x}^{0:T}, \mathbf{y}^{1:T}) = p(\mathbf{x}^0) \prod_{t=1}^{T} p(\mathbf{x}^t|\mathbf{y}^t, \mathbf{x}^{<t})p(\mathbf{y}^t|\mathbf{x}^{<t}). \tag{4}$$

We first specify the data likelihood term $p(\mathbf{x}^t|\mathbf{y}^t, \mathbf{x}^{<t})$, which we model autoregressively as a *Masked Autoregressive Flow* (MAF) (Papamakarios et al., 2017) applied to the frame sequence. This involves an autoregressive prediction network outputting $\mu_\phi$ and a scale parameter $\sigma$,

$$\mathbf{x}^t = \mu_\phi(\mathbf{x}^{<t}) + \sigma \odot \mathbf{y}_0^t \iff \mathbf{y}_0^t = \frac{\mathbf{x}^t - \mu_\phi(\mathbf{x}^{<t})}{\sigma}. \tag{5}$$

Conditioned on $\mathbf{y}_0^t$, this transformation is deterministic. The forward MAF transform $(\mathbf{y} \to \mathbf{x})$ converts the residual into the data sequence; the inverse transform $(\mathbf{x} \to \mathbf{y})$ decorrelates the sequence. The temporally decorrelated, sparse residuals $\mathbf{y}_0^{1:T}$ involve a simpler modeling task than generating the frames themselves. While the scale parameter $\sigma$ can also be conditioned on past frames, we did not find a benefit in practice.

The autoregressive transform in Equation 5 has also been adapted in a VAE model (Marino et al., 2021) as well as in neural video compression architectures (Agustsson et al., 2020; Yang et al., 2020; 2021a;b). These approaches separately compress latent variables that govern the next-frame prediction as well as frame residuals, thereby achieving state-of-the-art rate-distortion performance on high-resolution video content. While these works focused on compression, this paper focuses on generation.

We now specify the second factor in Eq. 4, the generative process of the residual variable, as

$$p_\theta(\mathbf{y}_{0:N}^t|\mathbf{x}^{<t}) = p(\mathbf{y}_N^t) \prod_{n=1}^N p_\theta(\mathbf{y}_{n-1}^t|\mathbf{y}_n^t, \mathbf{x}^{<t}). \tag{6}$$

We fix the top-level prior distribution to be a multivariate Gaussian with identity covariance. All other denoising factors are conditioned on past frames and involve prediction networks $M_\theta$,

$$p_\theta(\mathbf{y}_{n-1}|\mathbf{y}_n, \mathbf{x}^{<t}) = \mathcal{N}(\mathbf{y}_{n-1}|M_\theta(\mathbf{y}_n, n, \mathbf{x}^{<t}), \gamma\mathbf{I}). \tag{7}$$

As in Eq. 2, $\gamma$ is a hyperparameter. Our goal is to learn $\theta$.

**Inference Process.** Having specified the generative process, we next specify the inference process conditioned on the observed sequence $\mathbf{x}^{0:T}$:

$$q_\phi(\mathbf{y}_{0:N}^t|\mathbf{x}^{\leq t}) = q_\phi(\mathbf{y}_0^t|\mathbf{x}^{\leq t}) \prod_{n=1}^N q(\mathbf{y}_n^t|\mathbf{y}_{n-1}^t). \tag{8}$$

Since the residual noise is a deterministic function of the observed and predicted frame, the first factor is deterministic and can be expressed as $q_\phi(\mathbf{y}_0^t|\mathbf{x}^{\leq t}) = \delta(\mathbf{y}_0^t - \frac{\mathbf{x}^t - \mu_\phi(\mathbf{x}^{<t})}{\sigma})$. The remaining $N$ factors are identical to Eq. 1 with $\mathbf{x}_n$ being replaced by $\mathbf{y}_n$. Following Nichol & Dhariwal (2021), we use a cosine schedule to define the variance $\beta_n \in (0, 1)$. The architecture is shown in Figure 1b.

Equations 7 and 8 generalize and improve the previously proposed TimeGrad (Rasul et al., 2021) method. This approach showed promising performance in forecasting time series of comparatively smaller dimensions such as electricity prices or taxi trajectories and not video. Besides differences in architecture, this method neither models residuals nor considers the temporal dependency in posterior, which we identify as a crucial aspect to make the model competitive with strong VAE and GAN baselines (see Section 3.6 for an ablation).

**Optimization and Sampling.** In analogy to time-independent diffusion models, we can derive a variational lower bound that we can optimize using stochastic gradient descent. In analogy to to the derivation of Eq. 3 (Ho et al., 2020) and using the same definitions of $\bar{\alpha}_n$ and $\epsilon$, this results in

$$L(\theta, \phi) = \mathbb{E}_{\mathbf{x}, n, \epsilon} \sum_{t=1}^T ||\epsilon - f_\theta(\mathbf{y}_n^t(\phi), n, \mathbf{x}^{<t})||^2;$$

$$\mathbf{y}_n^t(\phi) = \sqrt{\bar{\alpha}_n}\mathbf{y}_0^t(\phi) + \sqrt{1 - \bar{\alpha}_n}\epsilon; \mathbf{y}_0^t(\phi) = \frac{\mathbf{x}^t - \mu_\phi(\mathbf{x}^{<t})}{\sigma}. \tag{9}$$

We can optimize this function using the reparameterization trick Kingma & Welling (2013), i.e., by randomly sampling $\epsilon$ and $n$, and taking stochastic gradients with respect to $\phi$ and $\theta$. For a practical scheme involving multiple time steps, we also employ teacher forcing Kolen & Kremer (2001). See Algorithm 1 for the detailed training and sampling procedure, where we abbreviated $f_{\theta,\phi}(\mathbf{y}_n^t, n, \mathbf{x}^{<t}) \equiv f_\theta(\mathbf{y}_n^t(\phi), n, \mathbf{x}^{<t})$.

---

**Algorithm 1:** Training (left) and Video Generation (right)

---

**while** *not converged* **do**
    Sample $\mathbf{x}^{0:T} \sim q(\mathbf{x}^{0:T})$;
    $n \sim \mathcal{U}(0, 1, 2, .., N)$;
    $L = 0$;
    **for** *t=1 to T* **do**
        $\epsilon \sim \mathcal{N}(\mathbf{0}, \mathbf{I})$;
        $\mathbf{y}_0^t = (\mathbf{x}^t - \mu_\phi(\mathbf{x}^{<t}))/\sigma$;
        $\mathbf{y}_n^t = \sqrt{\bar{\alpha}_n}\mathbf{y}_0^t + \sqrt{1 - \bar{\alpha}_n}\epsilon$;
        $L = L + ||\epsilon - f_{\theta,\phi}(\mathbf{y}_n^t, n, \mathbf{x}^{<t})||^2$
    **end**
    $(\theta, \phi) = (\theta, \phi) - \nabla_{\theta,\phi}L$
**end**

Get initial context frame $\mathbf{x}^0 \sim q(\mathbf{x}^0)$;
**for** *t=1 to T* **do**
    $\mathbf{y}_N^t \sim \mathcal{N}(\mathbf{0}, \mathbf{I})$;
    **for** *n=N to 1* **do**
        $\mathbf{z} \sim \mathcal{N}(\mathbf{0}, \mathbf{I})$;
        $\epsilon_\theta = f_{\theta,\phi}(\mathbf{y}_n^t, n, \mathbf{x}^{<t})$;
        $\tilde{\mathbf{y}}_{n-1}^t = \mathbf{y}_n^t - \frac{\beta_n}{\sqrt{1-\bar{\alpha}_n}}\epsilon_\theta$;
        $\mathbf{y}_{n-1}^t = \frac{1}{\sqrt{\alpha_n}}\tilde{\mathbf{y}}_{n-1}^t + \frac{1-\bar{\alpha}_{n-1}}{1-\bar{\alpha}_n}\beta_n\mathbf{z}$;
    **end**
    $\mathbf{x}^t = \sigma \odot \mathbf{y}_0^t + \mu_\phi(\mathbf{x}^{<t})$
**end**

---

## 3 Experiments

We compare Residual Video Diffusion (RVD) against five strong baselines, including three GAN-based models and two sequential VAEs. We consider four different video datasets and consider both probabilistic (CRPS) and perceptual (FVD, LPIPS) metrics, discussed below. Our model achieves a new state of the art in terms of perceptual quality while being comparable with or better than the best-performing sequential VAE in its frame forecasting ability.

### 3.1 Datasets

We consider four video datasets of varying complexities and resolutions. Among the simpler datasets of frame dimensions of $64 \times 64$, we consider the **BAIR** Robot Pushing arm dataset (Ebert et al., 2017) and **KTH Actions** (Schuldt et al., 2004). Amongst the high-resolution datasets (frame sizes of $128 \times 128$), we use **Cityscape** (Cordts et al., 2016), a dataset involving urban street scenes, and a two-dimensional **Simulation** dataset for turbulent flow of our own making that has been computed using the Lattice Boltzmann Method (Chirila, 2018). These datasets cover various complexities, resolutions, and types of dynamics.

**Preprocessing.** For KTH and BAIR, we preprocess the videos as commonly proposed (Denton & Fergus, 2018; Marino et al., 2021). For Cityscape, we download the portion titled *leftImg8bit_sequence_trainvaltest* from the official website[1]. Each video is a 30-frame sequence from which we randomly select a sub-sequence. All the videos are center-cropped and downsampled to 128x128. For the simulation dataset, we use an LBM solver to simulate the flow of a fluid (with pre-specified bulk and shear viscosity and rate of flow) interacting with a static object. We extract 10000 frames sampled every 128 ticks, using 8000 for training and 2000 for testing.

### 3.2 Training and Testing Details

The diffusion models are trained with 8 consecutive frames for all the datasets of which the first two frames as used as context frames. We set the batchsize to 4 for all high-resolution videos and to 8 for all low-resolution videos. The pixel values of all the video frames are normalized to $[-1, 1]$. The models are optimized using the Adam optimizer with an initial learning rate of $5 \times 10^{-5}$, which decays to $2 \times 10^{-5}$. All the models are trained on four NVIDIA RTX Titan GPUs in parallel for around 4-5 days. The number of diffusion depth is fixed to $N = 1600$ and the scale term is set to $\sigma = 2$. For testing, we use 4 context frames and predict 16 future frames for each video sequence. Wherever applicable, these frames are recursively generated. The model size is about 123 MegaBytes (32-bits float numbers) for high resolution models (128x128). To sample a 128x128 video frame, the model takes 18.5 seconds per 1000 iterations.

---

[1] https://www.cityscapes-dataset.com/downloads/

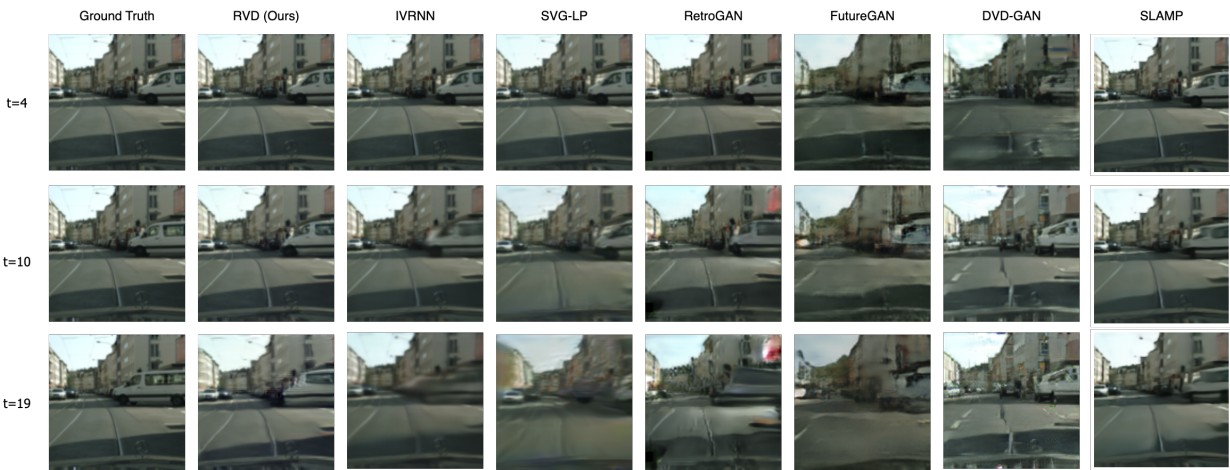

Figure 2: Generated frames on Cityscape (128x128). Compared to RVD (proposed), VAE-based models tend to become more blurry over time, while GAN-based methods generate artifacts and temporal inconsistencies.

### 3.3 Baseline Models

**SVG-LP** (Denton & Fergus, 2018) is an established sequential VAE baseline. It leverages recurrent architectures in all of encoder, decoder, and prior to capture the dynamics in videos. We adapt the official implementation from the authors while replacing all the LSTM with ConvLSTM layers, which helps the model scale to different video resolutions. **IVRNN** (Castrejon et al., 2019) is currently the state-of-the-art video-VAE model trained end-to-end from scratch. The model improves SVG by involving a hierarchy of latent variables. We use the official codebase to train the model. **SLAMP** (Akan et al., 2021) is a recent algorithm yielding stochastic predictions, similar to ours. It is also similar in spirit to our idea because it incorporates "motion history" to predict the dynamics for future frames. **FutureGAN** (Aigner & Körner, 2018) relies on an *encoder-decoder* GAN model that uses spatio-temporal 3D convolutions to process video tensors. In order to make the quality of the output more perceptually appealing, the paper employs the concept of progressively growing GANs. We use the official codebase to train the model. **Retrospective Cycle GAN** (Kwon & Park, 2019) employs a single generator that can predict both future and past frames given a context and enforces retrospective cycle constraints. Besides the usual discriminator that can identify fake frames, the method also introduces sequence discriminators to identify sequences containing the said fake frames. We used an available third-party implementation [2]. **DVD-GAN** (Clark et al., 2019) proposes an alternative dual-discriminator architecture for video generation on complex datasets. We also adapt a third-party implementation of the model to conduct our experiment [3].

### 3.4 Evaluation Metrics

We address two key aspects for determining the quality of generated sequences: perceptual quality and the models' probabilistic forecasting ability. For the former, we adopt FVD (Unterthiner et al., 2019) and LPIPS (Zhang et al., 2018), while the latter is evaluated using CRPS (Matheson & Winkler, 1976) to assess the marginal (pixel-based) predictions of future frames.

**Fréchet Video Distance** (FVD) compares sample realism by calculating 2-Wasserstein distance between the ground truth *video* distribution and the distribution defined by the generative model. Typically, an I3D network pretrained on an action-recognition dataset is used to capture low dimensional feature representations, the distributions of which are used in the metric. **Learned Perceptual Image Patch Similarity** (LPIPS), on the other hand, computes the $\ell_2$ distance between deep embeddings across all the layers of

---

[2]https://github.com/SaulZhang/Video_Prediction_ZOO/tree/master/RetrospectiveCycleGAN
[3]https://github.com/Harrypotterrrr/DVD-GAN

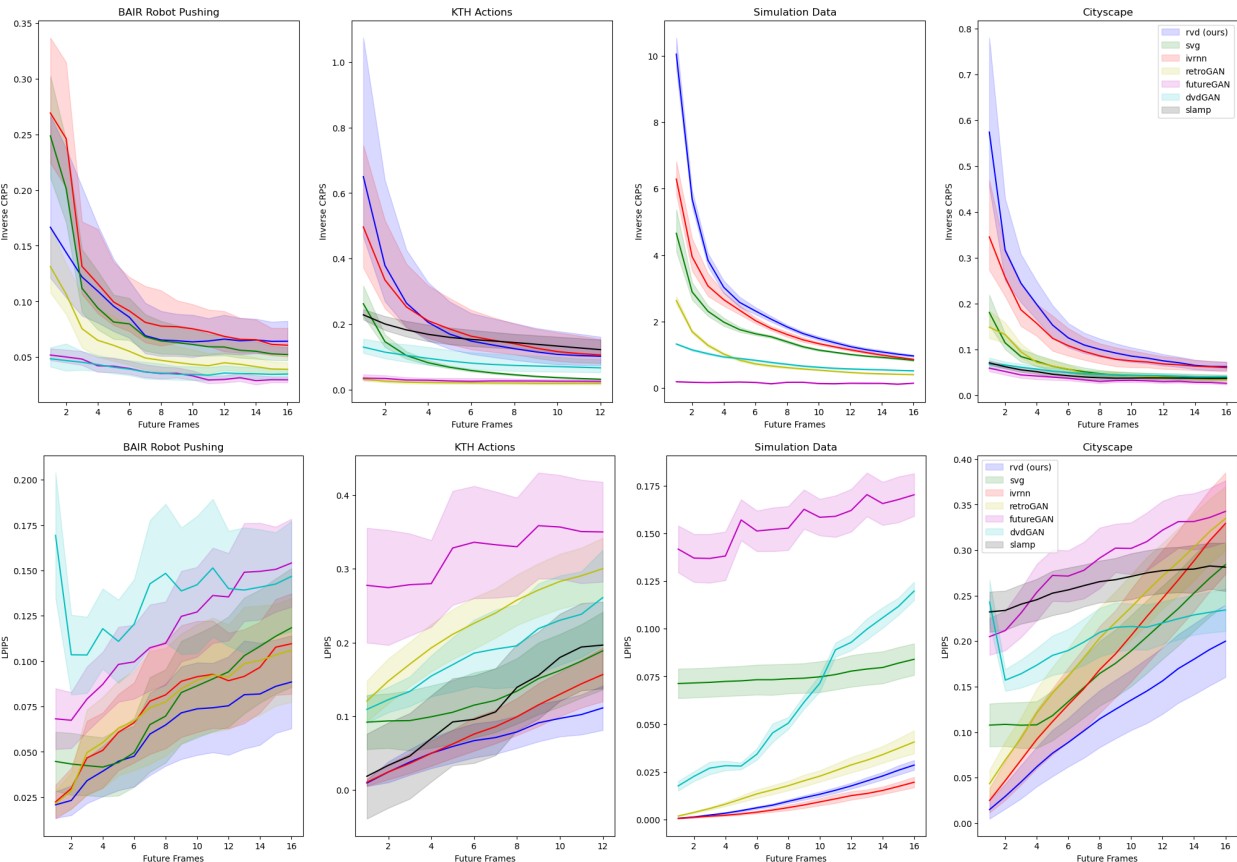

Figure 3: 1st row: Inverse CRPS scores (higher is better) as a function of the future frame index. The best performances are obtained by RVD (proposed) and IVRNN. Scores also monotonically decrease as the predictions worsen over time. 2nd row: LPIPS scores (Lower is better) show the per-frame-step perceptual quality, where the shaded region reflects the standard deviation of the sampled frames at the corresponding index.

a pretrained network which are then averaged spatially. The LPIPS score is calculated on the basis of individual frames and then averaged.

Another desirable property of video prediction methods is to reliably forecast future frames. Since ground truth videos exhibit multi-modal conditional distributions (e.g., a traffic light may switch from yellow to green or red), such multi-modality is best captures by proper Scoring Rules such as the Continuous Ranked Probability Score (CRPS). These metrics are frequently used in probabilistic forecasting problems in, e.g., meteorology or finance (Hersbach, 2000; Gneiting & Ranjan, 2011).

In a nutshell, CRPS compares a single-sample estimate of the ground truth CDF (a step function) with the model's CDF for the next frame. The latter can be efficiently estimated in one dimension by repeatedly sampling from the model. In expectation, CRPS not only rewards high accuracy of the mean prediction, but also good uncertainty estimates of the model. While CRPS is not commonly used in evaluating video prediction methods, we argue that it adds a valuable perspective on a model's uncertainty calibration.

### 3.5   Qualitative and Quantitative Analysis

Using the perceptual and probabilistic metrics mentioned above, we compare test set predictions of our video diffusion architecture against a wide range of baselines, which model underlying data density both explicitly and implicitly.

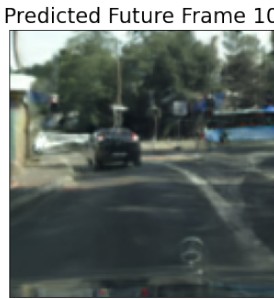
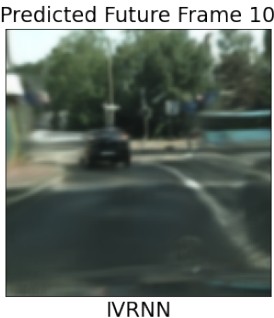
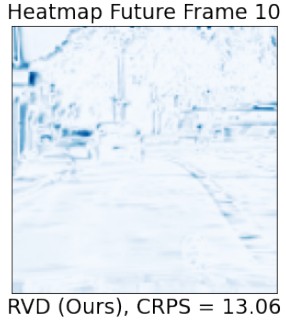
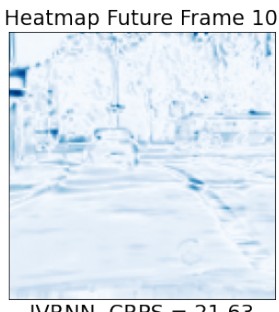

Figure 4: Spatially-resolved CRPS scores (right two plots, lower is better). We compare the performance of RVD (proposed) against IVRNN on predicting the $10^{th}$ future frame of a video from CityScape. Darker areas point to larger disagreements with respect to the ground truth.

| | **FVD↓** | | | | **LPIPS↓** | | | | **CRPS↓** | | | |
|---|---|---|---|---|---|---|---|---|---|---|---|---|
| | KTH | BAIR | Sim | City | KTH | BAIR | Sim | City | KTH | BAIR | Sim | City |
| RVD (ours) | **1351** | **1272** | **20** | 997 | **0.06** | 0.06 | 0.01 | **0.11** | 6.51 | 12.86 | **0.58** | **9.84** |
| IVRNN | 1375 | 1337 | 24 | 1234 | 0.08 | 0.07 | **0.008** | 0.18 | **6.17** | **11.74** | 0.65 | 11.00 |
| SLAMP | 1451 | N/A | N/A | 1853 | **0.05** | N/A | N/A | 0.23 | 6.18 | N/A | N/A | 23.6 |
| SVG-LP | 1783 | 1631 | 21 | 1465 | 0.12 | 0.08 | 0.01 | 0.20 | 18.24 | 13.96 | 0.75 | 19.34 |
| RetroGAN | 2503 | 2038 | 28 | 1769 | 0.28 | 0.07 | 0.02 | 0.20 | 27.49 | 19.42 | 1.60 | 20.13 |
| DVD-GAN | 2592 | 3097 | 147 | 2012 | 0.18 | 0.10 | 0.06 | 0.21 | 12.05 | 27.2 | 1.42 | 21.61 |
| FutureGAN | 4111 | 3297 | 319 | 5692 | 0.33 | 0.12 | 0.16 | 0.29 | 37.13 | 27.97 | 6.64 | 29.31 |

Table 1: Test set perceptual (FVD, LPIPS) and forecasting (CRPS) metrics, lower is better (see Section 3 for details).

Table 1 lists all the metric scores for our model and the baselines. Our model performs best in all cases in terms of FVD, a no-reference metric that measures frame quality irrespective of context and without reference to the ground truth. For LPIPS, a reference metric, our model also performs best in 3 out of 4 datasets. The perceptual performance is also verified visually in Figure 2, where RVD shows higher clarity on the generated frames and shows less blurriness in regions that are less predictable due to the fast motion.

We also reported CRPS scores in Table 1. Figure 3 shows 1/CRPS (higher is better) as a function of the frame index, revealing a monotonically decreasing trend along the time axis. This follows our intuition that long-term predictions become worse over time for all models. Our method performs best in 3 out of 4 cases. We can resolve this score also spatially in the images, as we do in Figure 4. Areas of distributional disagreement within a frame are shown in blue (right). See supplemental materials for the generated videos on other datasets.

### 3.6 Ablation Studies

We consider two ablations of our model. The first one studies the impact of applying the diffusion generative model for modeling residuals as opposed to directly predicting the next frames. The second ablation studies the impact of the number of frames that the model sees during training.

**Modeling Residuals vs. Dense Frames** Our proposed method uses a denoising diffusion generative model to generate *residuals* to a deterministic next-state prediction (see Figure 1b). A natural question arises whether this architecture is necessary or whether it could be simplified by directly generating the next frame $\mathbf{x}_0^t$ instead of the residual $\mathbf{y}_0^t$. To address this, we make the following adjustment. Since $\mathbf{y}_0^t$ and $\mathbf{x}_0^t$ have equal dimensions, the ablation can be realized by setting $\mu^t = 0$ and $\sigma = 1$. To distinguish from our proposed "Residual Video Diffusion" (RVD), we call this ablation "Video Diffusion" (VD). Note that this ablation can be considered a customized version of TimeGrad (Rasul et al., 2021) applied to video.

Table 2 shows the results. Across all perceptual metrics, the residual model performs better on all data sets. In terms of CRPS, VD performs slightly better on the simpler KTH and Bair datasets, but worse on the more complex Simulation and Cityscape data. We, therefore, confirm our earlier claims that modeling residuals over frames is crucial for obtaining better performance, especially on more complex high-resolution video.

**Frame Differences vs. Prediction Residuals** One may also wonder if similar results could have been obtained by modeling the residual relative to the last frame, as opposed to modeling the residual relative to a predicted frame. We called this ablation "SimpleRVD" (shown in Table 2), where we set $\mu_t = \hat{x}_{t-1}$. While the model can run more efficiently with this simplified scheme, Table 2 shows that the video quality is typically negatively affected. We conjecture that, since the predicted next frame may be already motion-compensated, the resulting residual is sparser and hence easier to capture by the diffusion model. Similar observations have been made in neural video compression (Agustsson et al., 2020; Yang et al., 2021b).

**Influence of Training Sequence Length** We train both our diffusion model and IVRNN on video sequences of varying lengths. As Table 2 reveals, we find that the diffusion model maintains a robust performance, showing only a small degradation on significantly shorter sequences. In contrast, IVRNN is more sensitive to the sequence length. We note that in most experiments, we outperform IVRNN even though we trained our model on shorter sequences. We also note that IVRNN leverages dense-connected hierarchical latent variables to capture long sequence dependency. Hence, optimizing the high-level latent variables can be challenging when the training sequence is not sufficiently long. While the diffusion model is also hierarchical, the model only learns a denoising mapping, which is much simpler scheme than dense-connected latents (as shown in Figure 1).

|  | FVD↓ | | | | LPIPS↓ | | | | CRPS↓ | | | |
|---|---|---|---|---|---|---|---|---|---|---|---|---|
|  | KTH | BAIR | Sim | City | KTH | BAIR | Sim | City | KTH | BAIR | Sim | City |
| VD(2+6) | 1523 | 1374 | 37 | 1321 | **0.066** | 0.066 | 0.014 | 0.127 | 6.10 | 12.75 | 0.68 | 13.42 |
| SimpleRVD(2+6) | 1532 | 1338 | 45 | 1824 | 0.065 | 0.063 | 0.024 | 0.163 | **5.80** | 12.98 | 0.65 | 10.78 |
| **RVD(2+6)** | **1351** | **1272** | **20** | **997** | **0.066** | **0.060** | 0.011 | **0.113** | 6.51 | 12.86 | **0.58** | **9.84** |
| RVD(2+3) | 1663 | 1381 | 33 | 1074 | 0.072 | 0.072 | 0.018 | **0.112** | 6.67 | 13.93 | 0.63 | 10.59 |
| IVRNN(4+8) | 1375 | 1337 | 24 | 1234 | 0.082 | 0.075 | **0.008** | 0.178 | **6.17** | **11.74** | 0.65 | 11.00 |
| IVRNN(4+4) | 2754 | 1508 | 150 | 3145 | 0.097 | 0.074 | 0.040 | 0.278 | 7.25 | 13.64 | 1.36 | 18.24 |

Table 2: Ablation studies on (1) modeling residuals (RVD, proposed) versus future frames (VD) and (2) training with different sequence lengths, where $(p + q)$ denotes $p$ context frames and $q$ future frames for prediction.

# 4 Related Work

Our paper combines ideas from video generation, diffusion probabilistic models, and neural video compression. As follows, we discuss related work along these lines.

## 4.1 Video Generation Models

Since the advent of modern deep learning, video generation and prediction has been a topic of ongoing interest; see (Oprea et al., 2020) and this paper's introduction. Videos can be generated based on side information of various types, such as images (Dorkenwald et al., 2021; Nam et al., 2019), text (Wu et al., 2021b; Singer et al., 2022; Gafni et al., 2022; Ramesh et al., 2022; Zhang et al., 2021; Zhou et al., 2022) or other videos (Wang et al., 2018). Alternatively, videos can also be generated unconditionally, e.g., from white noise (Clark et al., 2019; Saito et al., 2020; Yu et al., 2022). This survey focuses on conditional video generation, where one conditions the generation of future frames on a context of past frames.

Conditional video prediction is sometimes be treated as a supervised problem, where the focus is often on error metrics such as PSNR and SSIM (Lotter et al., 2017a; Byeon et al., 2018; Finn et al., 2016). Early works leverage deterministic methods to predict the most likely next-frames (Srivastava et al., 2015; Walker et al., 2015; Villegas et al., 2017; Lotter et al., 2017b; Liang et al., 2017; Finn et al., 2016). Generally

speaking, the downside of supervised approaches is that real-world videos display multi-modal behavior, i.e., the future is not uniquely predictable from the past (e.g., a traffic light may switch from yellow to red or green, a new object may or may not enter the scene etc). Treating video prediction as a supervised problem can therefore lead to mode averaging, perceived as blurriness.

Most recent methods focus on *stochastic generation* using deep generative models. In contrast to learning the average video dynamics, these methods try to match the conditional distribution of future frames given the past, typically by minimizing a divergence measure (as in GANs) or by optimizing a variational bound to a model's log likelihood (as in VAEs). Consequently, the evaluation has shifted to held-out likelihoods or perceptual metrics, such as FID or LPIPS. A large body of video generation research relies on variational deep sequential latent variable models (Babaeizadeh et al., 2018; Li & Mandt, 2018; Kumar et al., 2019; Unterthiner et al., 2018; Clark et al., 2019; Villegas et al., 2019; Babaeizadeh et al., 2021; Villegas et al., 2018; Yan et al., 2021; Rakhimov et al., 2020; Lee et al., 2021; Akan et al., 2021), to name a few. These works often draw on ealier works for modeling stochastic dynamics, e.g, Bayer & Osendorfer (2014); Chung et al. (2015), who included latent variables into recurrent neural networks. Later work (Denton & Fergus, 2018) extended the sequential VAE by incorporating more expressive priors conditioned on a longer frame context. IVRNN (Castrejon et al., 2019) further enhanced the generation quality by working with a hierarchy of latent variables, which to our knowledge is currently the best end-to-end trained sequential VAE model that can be further refined by greedy fine-tuning (Wu et al., 2021a). Normalizing flow-based models for video have been proposed while typically suffering from high demands on memory and compute (Kumar et al., 2019). Some works (Zhao et al., 2018; Franceschi et al., 2020; Marino, 2021; Marino et al., 2021) explored the use of residuals for improving video generation in sequential VAEs but did not achieve state of the art results.

Overall, the downside of VAE-based models is that they are trained to reconstruct the data. Blau & Michaeli (2018) theoretically showed that generative models are typically in conflict between data reconstruction tasks, and achieving a high-degree of *realism* (defined as matching the target distribution unconditionally, without artifacts). This suggests that VAEs may not be the final answer when it comes to video prediction.

Another line of sequential models rely on GANs (Vondrick et al., 2016b; Aigner & Körner, 2018; Kwon & Park, 2019; Yu et al., 2022; Tulyakov et al., 2018; Gui et al., 2021), which–at inference time–can be either deterministic and stochastic. In contrast to VAE-based models, these models tend to show fewer blurry artifacts. A downside of GANs is that that their loss function is mode-seeking (as opposed to mass covering), meaning that the data distribution is not covered at sufficient breadth, reflected in their typically worse performance in probabilistic distribution matching and forecasting metrics.

## 4.2 Diffusion Probabilistic Models

DDPMs have recently shown impressive performance on high-fidelity image generation. Sohl-Dickstein et al. (2015) first introduced and motivated this model class by drawing on a non-equilibrium thermodynamic perspective. Song & Ermon (2019) proposed a single-network model for score estimation, using annealed Langevin dynamics for sampling. Furthermore, Song et al. (2021c) used stochastic differential equations (related to diffusion processes) to train a network to transform random noise into the data distribution.

DDPM by Ho et al. (2020) is the first instance of a diffusion model scalable to high-resolution images. This work also showed the equivalence of DDPM and denoising score-matching methods described above. Subsequent work includes extensions of these models to image super-resolution (Saharia et al., 2021) or hybridizing these models with VAEs (Pandey et al., 2022). Apart from the traditional computer vision tasks, diffusion models were proven to be effective in audio synthesis (Chen et al., 2021; Kong et al., 2021), while Luo & Hu (2021) hybridized normalizing flows and diffusion model to generative 3D point cloud samples.

To the best of our knowledge, TimeGrad (Rasul et al., 2021) is the first sequential diffusion model for time-series forecasting. Their architecture was not designed for video but for traditional lower-dimensional correlated time-series datasets. Two concurrent preprints also study a video diffusion model (Ho et al., 2022; Voleti et al., 2022). Both works are based on alternative architectures and focuses primarily on perceptual metrics.

We will extend this survey in the camera-ready version.

### 4.3 Neural Video Compression Models

Video compression models typically employ frame prediction methods optimized for minimizing code length and distortion. In recent years, sequential generative models were proven to be effective on video compression tasks (Han et al., 2019; Yang et al., 2020; Agustsson et al., 2020; Yang et al., 2021a; Lu et al., 2019). Some of these models show impressive rate-distortion performance with hierarchical structures that separately encode the prediction and error residual. While compression models have different goals than generative models, both benefit from predictive sequential priors (Yang et al., 2021b). Note, however, that these models are ill-suited for generation since compression models typically have a small spatio-temporal context and are constructed to *preserve* local information, rather than to generalize (Yang et al., 2022).

## 5 Discussion

We proposed "Residual Video Diffusion": a new model for stochastic video generation based on denoising diffusion probabilistic models. Our approach uses a denoising process, conditioned on the context vector of a convolutional RNN, to generate a *residual* to a deterministic next-frame prediction. We showed that such residual prediction yields better results than directly predicting the next frame.

To benchmark our approach, we studied a variety of datasets of different degrees of complexity and pixel resolution, including Cityscape and a physics simulation dataset of turbulent flow. We compared our approach against two state-of-the-art VAE and three GAN baselines in terms of *both* perceptual and probabilistic forecasting metrics. Our method leads to a new state of the art in perceptual quality while being competitive with or better than state-of-the-art hierarchical VAE and GAN baselines in terms of probabilistic forecasting. Our results provide several promising directions and could improve world model-based RL approaches as well as neural video codecs.

**Limitations** The autoregressive setup of the proposed model allows conditional generation with at least one context frame pre-selected from the test dataset. In order to achieve unconditional generation of a complete video sequence, we need an auxiliary image generative model to sample the initial context frames. It's also worth mentioning that we only conduct the experiments on *single-domain* datasets with monotonic contents (e.g. Cityscape dataset only contains traffic video recorded by a camera installed in the front of the car), as training a large model for *multi-domain* datasets like Kinetics (Smaira et al., 2020) is demanding for our limited computing resources. Finally, diffusion probabilistic models tend to be slow in training which could be accelerated by incorporating DDIM sampling (Song et al., 2021a) or model distillation (Salimans & Ho, 2022).

**Potential Negative Impacts** Just as other generative models, video generation models pose the danger of being misused for generating deepfakes, running the risk of being used for spreading misinformation. Note, however, that a probabilistic video prediction model could also be used for anomaly detection (scoring anomalies by likelihood) and hence may help to detect such forgery.

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

## A  About CRPS

CRPS measures the agreement of a cumulative distribution function (CDF) $F$ with an observation $x$, $\mathrm{CRPS}(F, x) = \int_{\mathbb{R}} (F(z) - \mathbb{I}\{x \leq z\})^2 \, dz$, where $\mathbb{I}$ is the indicator function. In the context of our evaluation task, $F$ is the CDF of a single pixel within a single future frame assigned by the generative model. CRPS measures how well this distribution matches the empirical CDF of the data, approximated by a single observed sample. The involved integral can be well approximated by a finite sum since we are dealing with standard 8-bit frames. We approximate $F$ by an empirical CDF $\hat{F}(z) = \frac{1}{S} \sum_{s=1}^{S} \mathbb{I}\{X_s \leq z\}$; we stress that this does not require a likelihood model but only a set of $S$ stochastically generated samples $X_s \sim F$ from the model, enabling comparisons across methods.

## B  Architecture

Our architecture extends the previously proposed DDPM Ho et al. (2020) architecture to a temporally conditioned version. Figures 5 and 6 show the design of the proposed *denoising* and *transform* modules. Before describing these figures, we list important definitions and parameter choices below (see also Table 3 for more details). Codes are available at: `https://anonymous.4open.science/r/RVDC1FC`

- *Channel Dim* refers to the channel dimension of all the components in the first downsampling layer of the U-Net Ronneberger et al. (2015) style structure used in our approach.

- *Denoising/Transform Multipliers* are the channel dimension multipliers for subsequent downsampling layers (including the first layer) in the denoising/transform modules. The upsampling layer multipliers follow the reverse sequence.

- Each `ResBlock` He et al. (2016) leverages a standard implementation of the ResNet block with $3 \times 3$ kernel, LeakyReLU activation and Group Normalization.

- All `ConvGRU` Ballas et al. (2016) use a $3 \times 3$ kernel to deal with the temporal information.

- Each `LinearAttention` module involves 4 attention heads, each involving 16 dimensions.

- To condition our architecture on the denoising step $n$, we use positional encodings to encode $n$ and add this encoding to the `ResBlocks` (as in Figure 5).

- The `Upsample / Downsample` components in Figures 5 & 6 involve Deconvolutional / Convolutional networks that spatially scale the feature map with scaling factors 2 and 1/2, respectively.

Figure 5a shows the overall U-Net style architecture that has been adopted for denoising module. It predicts the noise information from the noisy residual (flowing through the blue arrows) at an arbitrary $n^{\text{th}}$ step (note that we perform a total of $N = 1600$ steps in our setup), conditioned on all the past context frames (flowing through the green arrows). The figure shows a low-resolution setting, where the number of downsampling and upsampling layers has been set to $L^{\text{denoise}} = 4$. Skip concatenations (shown as red arrows) are

performed between the `Linear Attention` of downsampling layer and first `ResBlock` of the corresponding upsampling layer as detailed in Figure 5b. Context conditioning is provided by a `ConvGRU` block within the downsampling layers that generates a context which is concatenated along the residual processing stream in the second `ResBlock` module. Additionally, each `ResBlock` module in either layers receives a positional encoding indicating the denoising step.

Figure 6a shows the U-Net style architecture that has been adopted for the transform module with the number of downsampling and upsampling layers $L^{\text{transform}} = 4$ in the low-resolution setting. Skip concatenations (shown as red arrows) are performed between the `ConvGRU` of downsampling layer and first `ResBlock` of the corresponding upsampling layer as detailed in Figure 6b.

| Video Resolutions | Channel Dim | Denoising Multipliers | Transform Multipliers |
|---|---|---|---|
| 64×64 | 48 | 1,2,4,8 ($L^{\text{denoise}} = 4$) | 1,2,2,4 ($L^{\text{transform}} = 4$) |
| 128×128 | 64 | 1,1,2,2,4,4 ($L^{\text{denoise}} = 6$) | 1,2,3,4 ($L^{\text{transform}} = 4$) |

Table 3: Configuration Table, see Section B for definitions.

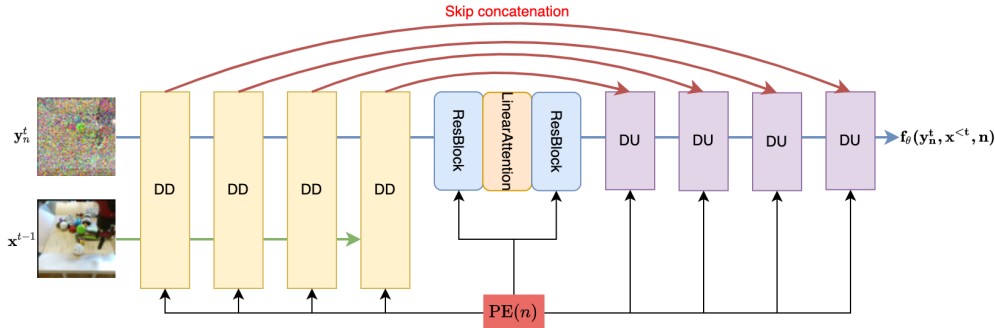

(a) Overview figure of the autoregressive denoising module using a U-net inspired architecture with skip-connections. We focus on the example of $L^{\text{denoise}} = 4$ downsampling and upsampling layers. Each of these layers, DD and DU, are explained in Figure b below. All DD and DU layers are furthermore conditioned on a positional encoding (PE) of the denoising step $n$.

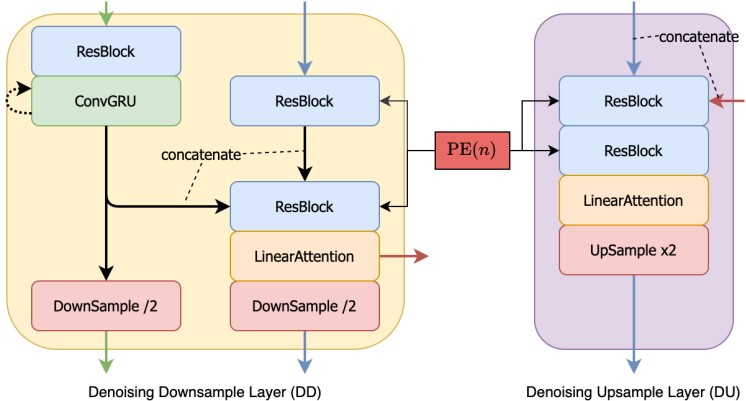

(b) Downsampling/Upsampling layer design for the autoregressive denoising module. Each arrow corresponds to the arrows with the same color in Figure (a). As in Figure (a), each residual block is conditioned on a positional encoding (PE) of the denoising step $n$.

Figure 5: Autoregressive denoising module.

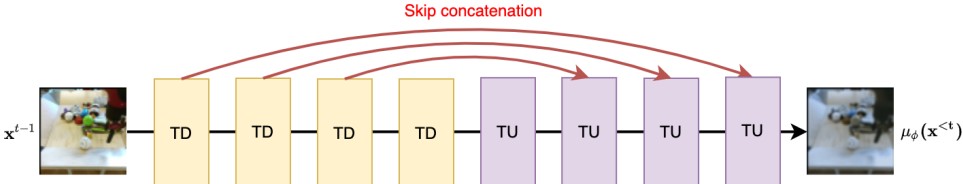

(a) Overview figure of the autoregressive transform module, predicting the mean next frame $\mu_t$. We use a U-net inspired architecture with an example size of $L^{\text{transform}} = 4$ upsampling and downsampling steps. Each arrow corresponds to the arrows with the same color in Figure (b). TD and TU layers are elaborated in Figure (b).

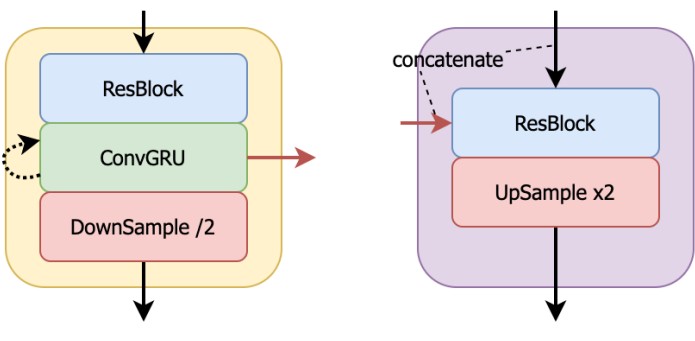

(b) Downsample/upsample layer design for autoregressive transform module. Each arrow corresponds to the arrows with the same color in Figure (a).

Figure 6: Autoregressive transform module for predicting the next frame $\mu_t$.

## C Deriving the Optimization Objective

The following derivation closely follows Ho et al. Ho et al. (2020) to derive the variational lower bound objective for our sequential generative model.

As discussed in the main paper, let $\mathbf{x}^{0:T}$ denote observed frames, and $\mathbf{y}_{0:N}^{1:T}$ the variables associated with the diffusion process. Among them, only $\mathbf{y}_{1:N}^{1:T}$ are the latent variables, while $\mathbf{y}_0^{1:T}$ are the the observed scaled residuals, given by $\mathbf{y}_0^t = \frac{\mathbf{x}^t - \mu_\phi(\mathbf{x}^{<t})}{\sigma}$ for $t = 1, ..., T$. The variational bound is as follows:

$$
\begin{aligned}
L &= \mathbb{E}_{\mathbf{x} \sim \mathcal{D}} \sum_{t=1}^{T} \mathbb{E}_{\mathbf{y}_{1..N}^t \sim q(\cdot | \mathbf{y}_0^t)} \left[ -\log \frac{p_\theta(\mathbf{y}_{0..N}^t | \mathbf{x}^{<t})}{q(\mathbf{y}_{1..N}^t | \mathbf{y}_0^t, \mathbf{x}^{<t})} \right] \\
&= \mathbb{E}_{\mathcal{D}} \sum_{t=1}^{T} \mathbb{E}_q \left[ -\log \frac{p(\mathbf{y}_N^t)}{q(\mathbf{y}_N^t | \mathbf{x}^{\leq t})} - \sum_{n>1}^{N} \log \frac{p_\theta(\mathbf{y}_{n-1}^t | \mathbf{y}_n^t, \mathbf{x}^{<t})}{q(\mathbf{y}_{n-1}^t | \mathbf{y}_n^t, \mathbf{x}^{\leq t})} \right. \\
&\quad \left. -\log p_\theta(\mathbf{y}_0^t | \mathbf{y}_1^t, \mathbf{x}^{<t}) \right].
\end{aligned}
\tag{10}
$$

The first term in Eq. 10, $-\log \frac{p(\mathbf{y}_N^t)}{q(\mathbf{y}_N^t | \mathbf{x}^{\leq t})}$, tries to match the $q(\mathbf{y}_N^t | \mathbf{x}^{\leq t})$ to the prior $p(\mathbf{y}_N^t) = \mathcal{N}(0, \mathbf{I})$. The prior has a fixed variance and is centered around zero, while $q(\mathbf{y}_N^t | \mathbf{x}^{\leq t}) = \mathcal{N}(\sqrt{\bar{\alpha}_n} \mathbf{y}_0^t, (1 - \bar{\alpha}_n) \mathbf{I})$. Because the variance of $q$ is also fixed, the only effect of the the first term is to pull $\sqrt{\bar{\alpha}_n} \mathbf{y}_0^t$ towards zero. However, in practice, $\bar{\alpha}_n = \prod_{i=0}^{N} (1 - \beta_i) \approx 0$, and hence the effect of this term is very small. For simplicity, therefore, we drop it.

To understand the third term in Eq. 10, we simplify

$$
\begin{aligned}
&- \log p_\theta(\mathbf{y}_0^t | \mathbf{y}_1^t, \mathbf{x}^{<t}) \\
&= \frac{1}{2\sigma\gamma_1} ||\mathbf{x}^t - \mu(\mathbf{x}^{<t}) - \sigma M_\theta(\mathbf{y}_1^t, \mathbf{x}^{<t}, 1)||^2 + const.,
\end{aligned}
\tag{11}
$$

Eq. 11 suggests that the third term matches the diffusion model's output to the frame residual, which is also a special case of $L^{\mathrm{mid}}$ we elaborate below.

Deriving a parameterization for the second term of Eq. 10, $L^{\mathrm{mid}} := -\sum_{n>1}^N \log \frac{p_\theta(\mathbf{y}_{n-1}^t | \mathbf{y}_n^t, \mathbf{x}^{<t})}{q(\mathbf{y}_{n-1}^t | \mathbf{y}_n^t, \mathbf{x}^{\leq t})}$, we recognize that it is a KL divergence between two Gaussians with fixed variances:

$$
q(\mathbf{y}_{n-1}^t | \mathbf{y}_n^t, \mathbf{x}^{\leq t}) = q(\mathbf{y}_{n-1}^t | \mathbf{y}_n^t, \mathbf{y}_0^t) = \mathcal{N}(\mathbf{y}_{n-1}^t; M(\mathbf{y}_n^t, \mathbf{y}_0^t), \gamma_n \mathbf{I}),
\tag{12}
$$

$$
p_\theta(\mathbf{y}_{n-1}^t | \mathbf{y}_n^t, \mathbf{x}^{<t}) = \mathcal{N}(\mathbf{y}_{n-1}^t; M_\theta(\mathbf{y}_n^t, \mathbf{x}^{<t}, n), \gamma_n \mathbf{I}).
\tag{13}
$$

We define $M(\mathbf{y}_n^t, \mathbf{y}_n^0) = \frac{\sqrt{\bar{\alpha}_{n-1}}\beta_n}{1-\bar{\alpha}_n}\mathbf{y}_0^t + \frac{\sqrt{1-\beta_n}(1-\bar{\alpha}_{n-1})}{1-\bar{\alpha}_n}\mathbf{y}_n^t$. The KL divergence can therefore be simplified as the L2 distance between the means of these two Gaussians:

$$
L_n^{\mathrm{mid}} = \frac{1}{2\gamma_n} ||M(\mathbf{y}_n^t, \mathbf{y}_n^0) - M_\theta(\mathbf{y}_n^t, \mathbf{x}^{<t}, n)||^2
\tag{14}
$$

As $\mathbf{y}_n^t$ always has a closed form when $\mathbf{y}_0^t$ is given: $\mathbf{y}_n^t = \sqrt{\bar{\alpha}_n}\mathbf{y}_0^t + \sqrt{1-\bar{\alpha}_n}\epsilon$ (See Section 3.1), we parameterize Eq 14 to the following form:

$$
L_n^{\mathrm{mid}} = \frac{1}{2\gamma_n} ||\frac{1}{\sqrt{1-\beta_n}}(\mathbf{y}_n^t - \frac{\beta_n}{\sqrt{1-\bar{\alpha}_n}}\epsilon) - M_\theta(\mathbf{y}_n^t, \mathbf{x}^{<t}, n)||^2.
\tag{15}
$$

It becomes apparent that $M_\theta$ is trying to predict $\frac{1}{\sqrt{1-\beta_n}}(\mathbf{y}_n^t - \frac{\beta_n}{\sqrt{1-\bar{\alpha}_n}}\epsilon)$. Equivalently, we can therefore predict $\epsilon$. To this end, we replace the parameterization $M_\theta$ by the following parameterization involving $f_\theta$:

$$
\begin{aligned}
&L_n^{\mathrm{mid}} \\
&= \frac{1}{2\gamma_n} ||\frac{1}{\sqrt{1-\beta_n}}(\mathbf{y}_n^t - \frac{\beta_n}{\sqrt{1-\bar{\alpha}_n}}\epsilon) \\
&\quad - \frac{1}{\sqrt{1-\beta_n}}(\mathbf{y}_n^t - \frac{\beta_n}{\sqrt{1-\bar{\alpha}_n}}f_\theta(\mathbf{y}_n^t, \mathbf{x}^{<t}, n))||^2
\end{aligned}
\tag{16}
$$

The resulting stochastic objective can be simplified by dropping all untrainable parameters, as suggested in Ho et al. (2020). As a result, one obtains the simplified objective

$$
L_n^{\mathrm{mid}} \equiv ||\epsilon - f_\theta(\mathbf{y}_n^t(\phi), n, \mathbf{x}^{<t})||^2,
\tag{17}
$$

which is the denoising score matching objective revealed in Eq. 9 of our paper.

# D   Additional Generated Samples

In this section, we present some qualitative results from our study for some additional datasets apart from CityScape. More specifically, we present some examples from our Simulation data, BAIR Robot Pushing data and KTH Actions data. In the following figures, the top row consists of ground truth frames, both contextual and predictive, while the subsequent rows are the generated frames from our method and some other VAE and GAN baselines. It is quite evident that our our method and IVRNN are the strongest contenders. Perceptually, the FVD metric indicates that we outperform on every dataset, whereas statistically, the CRPS metric shows our top performance on high-resolution datasets and a competitive performance against IVRNN on the other two datasets.

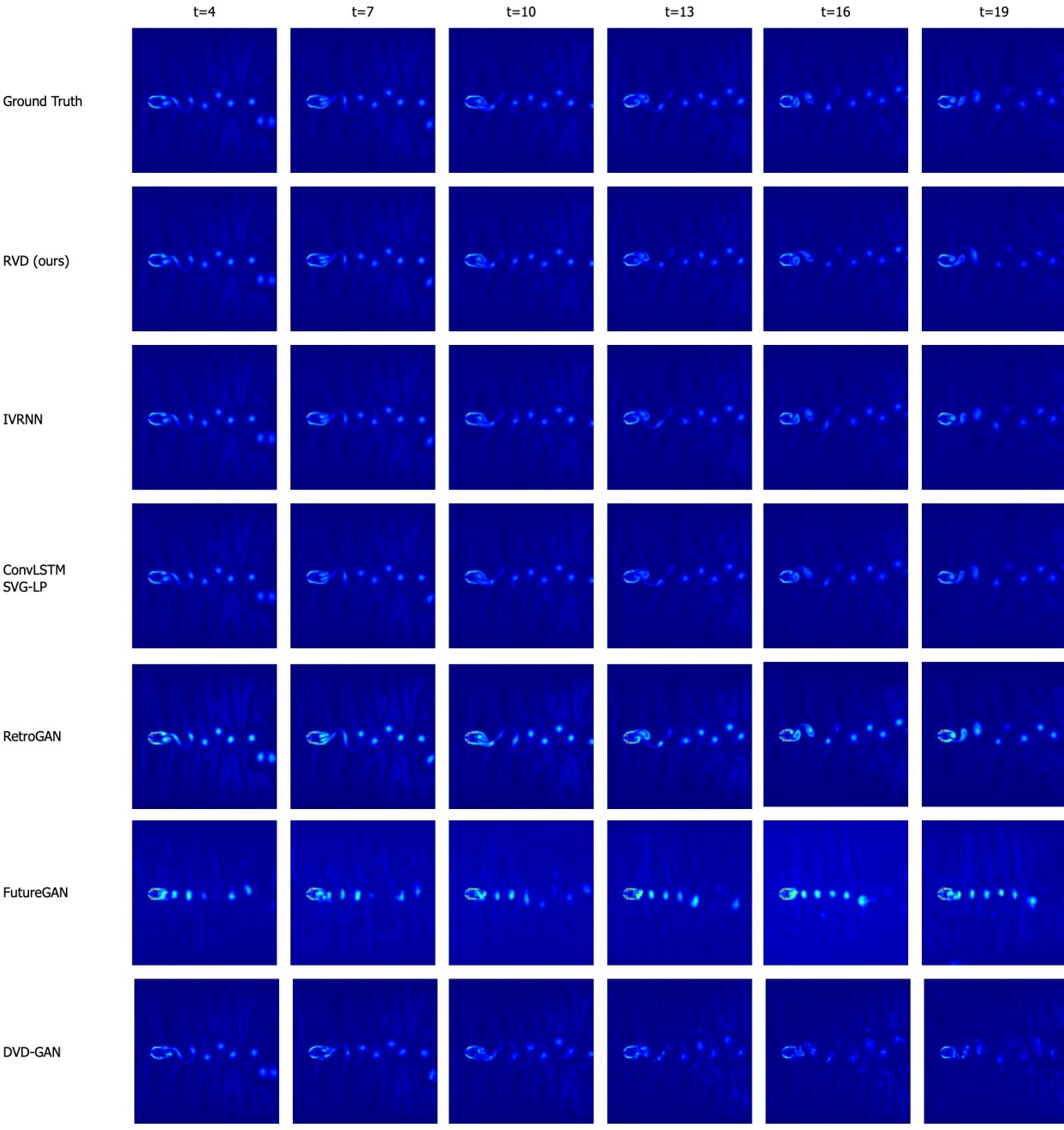

Figure 7: Prediction Quality for Simulation data. Top row indicates the ground truth, wherein we feed 4 frames as context (from $t = 0$ to $t = 3$) and predict the next 16 frames (from $t = 4$ to $t = 19$). This is a high resolution dataset ($128 \times 128$) which we generated using a Lattice Boltzmann Solver. It simulates the von Kármán vortex street using Navier-Stokes equations. A fluid (with pre-specified viscosity) flows through a 2D plane interacting with a circular obstacle placed at the center left. This leads to the formation of a repeating pattern of swirling vortices, caused by a process known as vortex shedding, which is responsible for the unsteady separation of flow of a fluid around blunt bodies. Colors indicate the vorticity of the solution to the simulation.

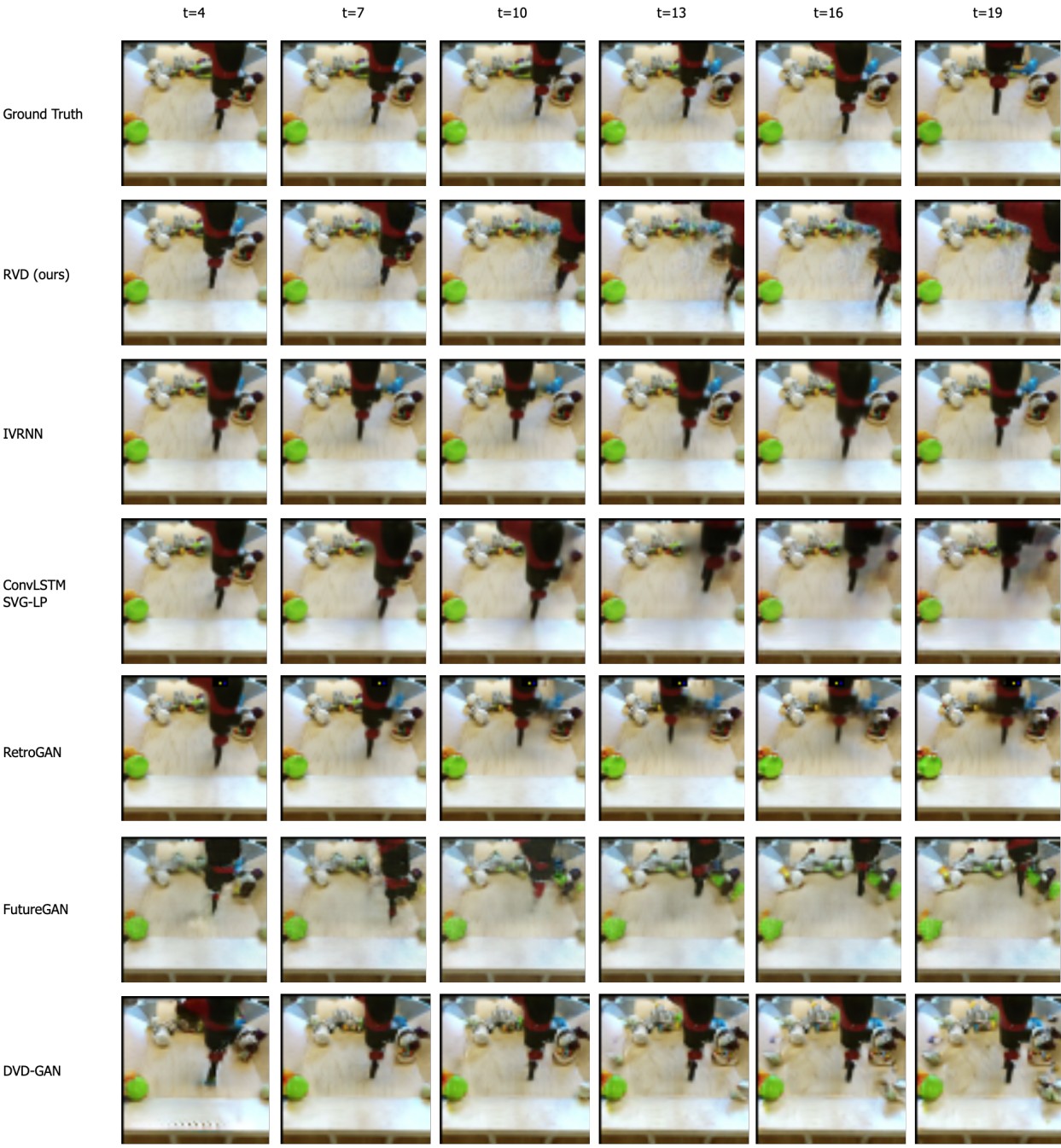

Figure 8: Prediction Quality for BAIR Robot Pushing. As seen before, the top row indicates the ground truth, wherein we feed 4 frames as context (from $t = 0$ to $t = 3$) and predict the next 16 frames (from $t = 4$ to $t = 19$). This is a low resolution dataset ($64 \times 64$) that captures the motion of a robotic hand as it manipulates multiple objects. Temporal consistency and occlusion handling are some of the big challenges for this dataset.

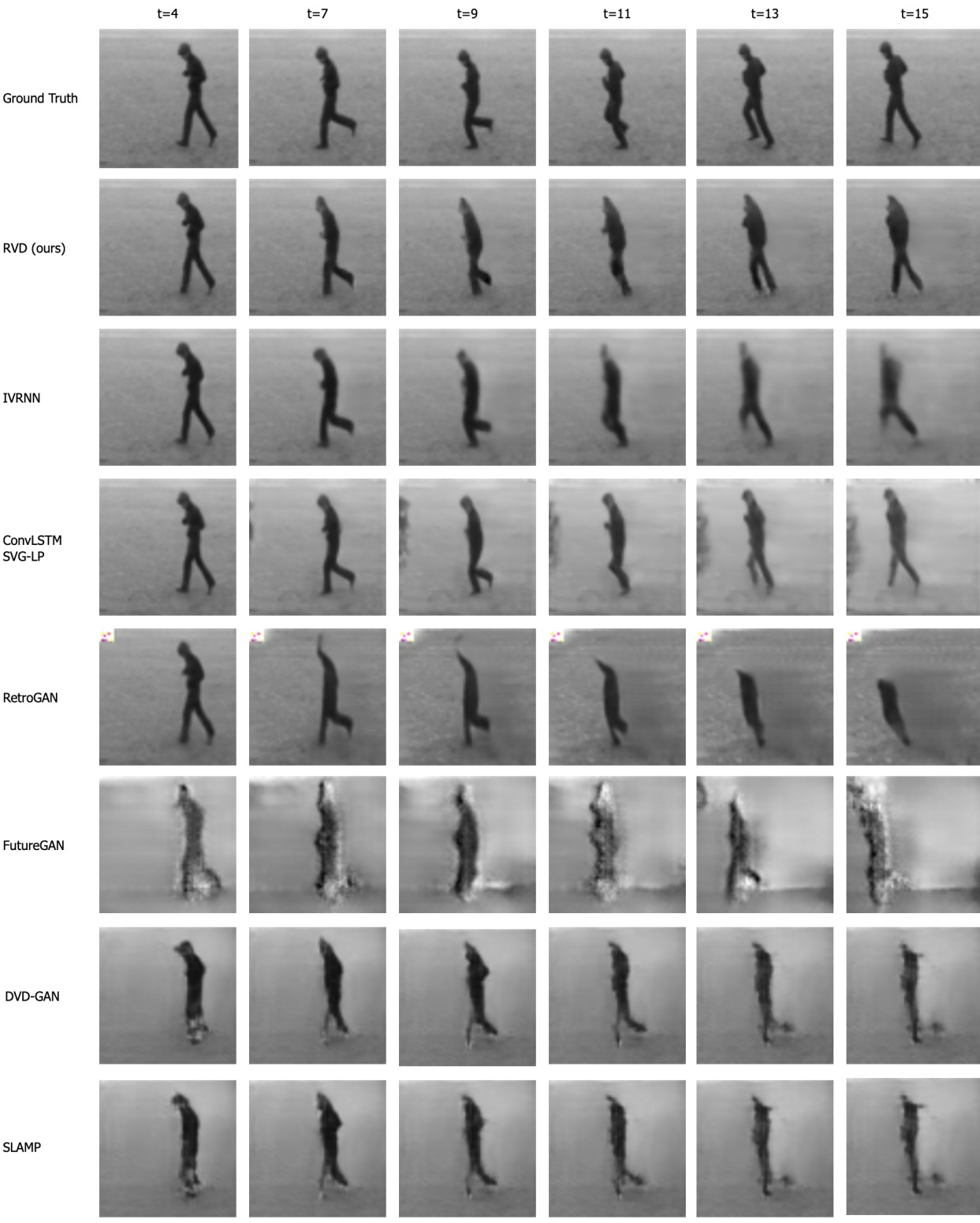

Figure 9: Prediction Quality for KTH Actions. As seen before, the top row indicates the ground truth, wherein we feed 4 frames as context (from $t = 0$ to $t = 3$), but unlike other datasets, we only predict the next 12 frames (from $t = 4$ to $t = 15$).

