# OpenReview forum: "Diffusion Probabilistic Modeling for Video Generation"
_TMLR — Rejected by TMLR_

### Review · Reviewer_11nc · 2022-12-16

**Summary Of Contributions:**

This paper proposes a diffusion model to tackle video generation. The model works by predicting the next frame using a convolutional recurrent neural network, and refining the prediction with a conditional diffusion model. This combination enables their approach to scale well and achieve crisp generation results. The paper also proposes an adaption of the Continuous Ranked Probability Score for video prediction.

**Audience:**

Yes

**Broader Impact Concerns:**

The model is trained on datasets not containing people and I do not think there are significant concerns other than the ones already discussed within the paper.

**Claims And Evidence:**

Yes

**Requested Changes:**

- Since a new metric (CRPS for videos) is proposed, further evaluation or justification is necessary to show that it is indeed useful for the video prediction setting. In particular, I would be more convinced by human evaluation results showing that CRPS is better correlated with human judgment (over existing metrics like FVD), or robust evaluations (either human evaluations or statistical derivation) showing that CRPS captures certain details that FVD misses.
- Can you provide more details about the efficiency and training requirements? For example, some details that would be useful for readers are: how many parameters does the model have, how long does a single model take to train, how much GPU memory is required, and what is the inference throughput?
- Does the model scale well to higher resolutions? For example, is it possible to show some results on 256px or even 512px videos? If not, is this because the model requires greater memory or compute requirements compared to baseline models?


**Strengths And Weaknesses:**

In general, I think that this paper is a very nice contribution for the video generation and diffusion modeling communities. It effectively shows how to combine a recurrent neural network with diffusion models, and that both components have complementary benefits. My main concern is with the proposed CRPS metric – if it is to be a useful metric for measuring video generation moving forward, it is necessary to show that it is useful over FVD/LPIPS.

## Strengths:
The paper is well written, easy to follow, and provides sufficient mathematical rigor and background for explaining the proposed model and evaluation metrics.
The design choices are generally well supported by the ablation experiments. In particular, the proposed diffusion model for prediction refinement improves generation quality (Table 2).
The proposed model is simple and appears to be easy to train (they train it on a single GPU).

## Weaknesses:

The effectiveness of the proposed CRPS metric is unclear. Usually when a new metric is proposed, human evaluations are necessary to determine if it has strong correlation with human judgment. In particular, since one of the contributions of the paper is the CRPS metric for generated video sequences, it is essential to show that the proposed metric is either (1) better correlated with human judgment compared to FVD or LPIPS, or (2) concretely show (through statistics or human evaluations) that it captures certain aspects that other metrics do not.
The paper is missing references and discussion to relevant literature on video diffusion models [1, 2] and more recent hierarchical video generation models [3].


## References
[1] Höppe, Tobias, et al. "Diffusion models for video prediction and infilling." Transactions on Machine Learning Research, (2022).
[2] Ho, Jonathan, et al. "Imagen video: High definition video generation with diffusion models." arXiv preprint arXiv:2210.02303 (2022).
[3] Lee, Wonkwang, et al. "Revisiting hierarchical approach for persistent long-term video prediction.", ICLR, (2021).

---

> ### Author Response · Authors · 2023-02-18
> **Individual Response**
>
> > missing references [...] to relevant literature on video diffusion models and more recent hierarchical video generation models.
>
> Thanks for pointing out references to related works; we extended our related work discussion on video generative models. Please see our general comments on other video diffusion models and reasons for why we decided against direct comparisons. We will add these references and discussions once our paper will get accepted.
>
> > It is necessary to show that [CRPS] it is useful over FVD/LPIPS
>
> We acknowledge that our presentation about CRPS can be improved. However, CRPS is *not* a new metric: it’s an established metric for measuring the agreement between a predictive distribution and observations, and it is widely used in traditional time series forecasting. As such, it is a quantitative forecasting metric which complements the otherwise perceptual evaluation based on FVD and LPIPS (see the Wikipedia article on “Scoring Rule”).
>
> We will de-emphasize marginal CRPS as a contribution and will extend Fig. 3 to also show LPIPS for future frames.
>
> > concretely show (through statistics or human evaluations) that [CRPS] captures certain aspects that other metrics do not
>
> CRPS is an established summary measure for the evaluations of probabilistic predictions (see “Scoring Rule” article on Wikipedia).  As such, it is complementary to perceptual reference (e.g., LPIPS) or no-reference metrics (e.g., FVD).
>
> > details about the efficiency and training requirements
>
> We appreciate the suggestions. We added these details in Section 3.2.
>
> > Does the model scale well to higher resolutions?
>
> Yes, our method scales linearly in the frame dimensions. Our model is fully convolutional. However, the diffusion model in its current form is computationally expensive due to the iterative denoising procedure. Due to our limited computational resources, we restricted our experiments to 128x128 and 64x64 videos.

---

### Review · Reviewer_z5Zc · 2022-12-22

**Summary Of Contributions:**

This paper proposes a method for video prediction that combines a deterministic module with a diffusion based stochastic module. The main motivation is that diffusion methods alone cannot perform accurate prediction, and that it is better to simply apply refinement using diffusion on pixels generated deterministically. This paper also proposes a method called CRPS that is applicable to video generation. In experiments, the authors compare favorably against the chosen baselines and ablations of their own method using standard metrics and their newly proposed metric.

**Audience:**

No

**Broader Impact Concerns:**

No concerns about broader impact.

**Claims And Evidence:**

No

**Requested Changes:**

In conclusion, this paper has critical issues in terms of 1) the claims that have already been shown not to be true by pure diffusion methods, 2) Seemingly lack of familiarity with the latest works in video prediction and 3) potential lack of understanding of what we need video prediction methods to reflect in the generated/predicted frames.

**Strengths And Weaknesses:**

Weaknesses:
Missing baselines, and related works:
- All the baselines used in this paper are from 3-4 years ago while the paper makes SOTA claims in video prediction. There have been methods like FitVid (https://arxiv.org/abs/2106.13195), BigSVG (https://arxiv.org/abs/1911.01655) which have numbers in popular / challenging benchmarks, thus, it's just a matter of running experiments on those benchmarks.

- More recently, diffusion based video prediction/generation has already been explore and contradict the claims of this paper. Methods such as video diffusion from Ho et al., 2021 (cited in this paper) came out 9 months before this submission is called "concurrent work", and MCVD (https://arxiv.org/abs/2205.09853)  which came out 2 months before this submission perform diffusion directly in pixel space successfully completely contradicting the main premise of this submission. I suggest the authors revise this submission and either perform experiments on the same datasets as the diffusion works mentioned above or run experiments on the datasets used in this submission (there is official code available online for MCVD and unofficial implementations of video diffusion also available online). I suggest the authors to also look at the video prediction numbers from other papers provided in these papers as they are the current SOTA methods.

- Finally, there are many video prediction/generation works ignored in the related works such as MCNet (https://arxiv.org/abs/1706.08033), PredNet (https://arxiv.org/abs/1605.08104), DrNet (https://arxiv.org/abs/1705.10915).

CRPS doesn't make much sense:
- By now, there has been a consensus in video prediction / generation work that pixel-level metrics tend to provide inaccurate evaluations regarding the plausibility of the predicted future. Also, slight pixel variations that do not change the semantics of the predicted future are heavily penalized. This is the reason why metrics such as MSE, PSNR, SSIM, and other pixel-level metrics are not being used as much and authors resource to perceptual metrics which contain a more semantically meaningful evaluation which is what we care about when using the predicted frames for other downstream tasks. CRPS as I understand it is a pixel-level evaluation which will suffer from the same issues the pixel level evaluations above suffer from. Is it possible to apply this metric to some more semantically meaningful inputs?

---

> ### Author Response · Authors · 2023-02-18
> **Individual Response**
>
> > Missing baselines, and related works
>
> We appreciate the reviewer pointing out baselines that we have missed. Since video prediction is a very resource-hungry field of research, we needed to make choices and felt that our previous baselines were above average in quantity; however, we agreed to the need to add a more recent baseline (SLAMP, from ICCV 2021) to our results. We will discuss the remaining non-diffusion-based baselines in the related work.
>
> In terms of novelty, please see our general response. The Action Editor will be able to support our claim on concurrency with VDM by Ho et al, 2022.
>
> > This paper also proposes a method called CRPS
>
> > CRPS doesn’t make much sense
>
> Note that CRPS is *not* a new metric; it is a widely-used scoring rule in statistics for probabilistic forecasting (see “Scoring Rule“ on Wikipedia). However, we do acknowledge that we have over-emphasized the use of CRPS for video frame prediction. In our revised version, we treat CRPS more on par with the two perceptual metrics, FID and LPIPS. We also no longer claim marginal CRPS as a contribution. CRPS is a useful metric in that it is a proper scoring rule for probabilistic forecasting; hence it complements perceptual scores.

---

### Review · Reviewer_eJfF · 2023-02-01

**Summary Of Contributions:**

The paper suggests a two-stage approach for video generation called RVD: (1) predict the next frame with autoregressive model and (2) revise the residual with a diffusion model. The paper also applies the CRPS metric for videos.

**Audience:**

Yes

**Broader Impact Concerns:**

No.

**Claims And Evidence:**

Yes

**Requested Changes:**

Address the concerns [W1-4] above.

**Strengths And Weaknesses:**

**Strength**
- Applying diffusion for video generation is a timely and important topic.
- The paper aims to design video-specific components instead of naively applying diffusion.



**Weakness**

[W1] Inefficient design of RVD.
- RVD has two computationally heavy components: (1) autoregressive generation of $T$ frames and (2) iterative refinement of $N$ diffusion steps. Thus, RVD takes $T \times N$ times of iterations, unlike most other approaches need either $T$ or $N$. It is way too expensive compared to the competitors.
- The autoregressive model also may be unnecessary. Since the subsequent video frames have many redundancies, many prior works leveraged this property by only predicting the residual between the next and current frames $x_t - x_{t-1}$. Since RVD needs the correction step anyway, why don't just generate $x_t - x_{t-1}$ with the diffusion model? It drops an expensive ad-hoc RNN model and could reduce error accumulation.
- Thus, I think the current version of RVD is not mature in both technical (heavy computation) and domain-specific (not leveraging the property of video well) perspectives.


[W2] Missing comparison with recent methods.
- The paper only compares with methods before 2019, which is too outdated for the late 2022 submission. There are tons of video generation works after 2019, like DIGAN [1] or Video Diffusion [2] to name a few. Also, there are tons of video generation works (though many of them are on text-to-video generation) such as Imagen Video or Make-A-Video, which could be at least discussed in the related work section.

[W3] Misleading presentation of CRPS.
- The paper is just averaging the CRPS metric (for a single pixel) over spatial and temporal axes for videos, which is a trivial extension. However, the paper states it as a contribution in the abstract and introduction, which can confuse the authors about the amount of this paper's contribution.
- Furthermore, the paper introduces CRPS as a metric to check if "generated sequence covers the full distribution of possible outcomes." However, to my best knowledge, CRPS becomes optimal when every probabilistic prediction collapses to a single GT value. Thus, it is just one way to compute MSE, not a metric to evaluate the diversity (or cover-ness of the full distribution) of multiple outputs.


[W4] Limited in-depth discussions.
- The paper contains only two main messages: (1) a main table Tab. 1 showing that RVD surpasses baselines in 4 datasets (Fig. 2 and 3 also say the same message), (2) an ablation study Tab. 2 suggesting that 2a) creating residual is easier than creating the entire frame $x_t$ (but not compared with creating $x_t - x_{t-1}$ I suggested above) and 2b) RVD is less sensitive to the sequence length than IVRNN (but without a proper explanation). As an academic paper, the discussion should be more than just saying our method is SOTA (which is not very surprising at the cost of diffusion for refinement) so that the readers can learn some new things from the paper. In the current form, I wonder if the contents are enough to be a full conference/journal paper.
- Many possible directions can make the discussion more plentiful. For example, understanding why RVD is more robust to the sequence length may lead to a better method design.
- If that is too challenging, there are also many possible contents that the paper can easily include following the prior works (e.g., see the experiments of Video Diffusion [2]). For example, one can extend uncond. video generation for video prediction (or other conditional generation) by applying classifier guidance. Also, the paper could present the sampling time of RVD and explore ways to reduce the cost (e.g., apply DDIM).


[1] Generating Videos with Dynamics-aware Implicit Generative Adversarial Networks. ICLR, 2022.\
[2] Ho et al. Video Diffusion Models. arXiv, 7 Apr 2022.

---

> ### Author Response · Authors · 2023-02-18
> **Individual Response**
>
> > Inefficient design of RVD.
>
> We acknowledge that the diffusion model can be computationally inefficient due to the iterative denoising steps, which can result in long processing times. While other generative models such as GANs or VAEs can generate videos autoregressively in T iterations (where T is the video length), autoregressive diffusion models will encounter a multiplicative factor N, i.e., the number of denoising steps. While Ho et. al. (2022) leverage 3D convolutions, note that their approach formally still scales as T*N (since T is one axis of the data tensor). While their model can parallelize the generation for a k-frames window, they still need an autoregressive procedure to produce longer videos.
>
> > Missing comparison with recent methods.
>
> We appreciate the many constructive suggestions. To prove competitive performance against newer methods, we added another recent video prediction baseline from ICCV 2021 called Stochastic Latent Appearance and Motion Prediction (SLAMP). We chose this baseline both because it is recent and the algorithm yields stochastic predictions, similar to ours. It is also similar in spirit to our idea because it incorporates “motion history” to predict the dynamics for future frames. At this point, we have tested it on KTH and CityScape datasets. For Cityscape, SLAMP yielded an FVD score of 1853 and LPIPS score of 0.238, both worse than the ones achieved by the proposed RVD. For KTH Actions dataset, the algorithm achieved an FVD score of 1451 while an LPIPS score of 0.05. We will add all final results in the final paper version.
>
> > Misleading presentation of CRPS.
>
> Please also refer to our general response. We generally acknowledge the fact that CRPS is a well-known metric, and we will move some of its presentation to the appendix to avoid the wrong impression that we invented the metric. In terms of suitability for video prediction, we emphasize that CRPS is just one of the three metrics we used; and we will make sure to add additional evaluations for FID and LPIPS wherever we use CRPS. We still like to keep CRPS in our paper since it is the only probabilistic forecasting metric evaluated.
>
> Furthermore, you wrote that “to my best knowledge, CRPS becomes optimal when every probabilistic prediction collapses to a single GT value”. This is true (it is evaluated on individual samples and not on a distribution level) and our previous description was misleading, and we will correct it in this regard. However, CRPS is not “just one way to compute MSE”. Rather, CRPS is a proper scoring rule typically used in probabilistic forecasting problems (see Wikipedia article on “Scoring Rules”). As such, it does take the possible multi-modality of a prediction into account, unlike MSE. We will add a more nuanced discussion in our paper.
>
>
> > Limited in-depth discussions.
>
> We added a discussion on why IVRNN is more sensitive to training sequence length in our paper. The primary reason could be that IVRNN uses hierarchical latent variables, where all the latent variables are densely connected. In this case, the model needs a longer training sequence to fully exploit the hierarchical features of the videos.
>
> We also added an ablation study about using naive residual $x_t-x_{t-1} $(SimpleRVD) vs. RVD. The results show that predicting the next frame indeed leads to a performance improvement. We note that even in the absence of a predicted next frame, our approach would need an autoregressive module to condition the residual diffusion module on a long-range temporal context (otherwise, the video would literally diffuse over time). Our model architecture was inspired by recent architectures from neural video compression, which exploits a 2-stage video reconstruction process (prediction+residual reconstruction) to minimize the bitrate usage. In our method, the deterministic module will generate a “rough” prediction and the diffusion model stochastically generates the details. We will add these details to the paper.
>
> We generally agree that scientific papers benefit from extended discussions and ablations, and we hope that the additional ablations and comparisons will strengthen our paper. Note that for developing video prediction models, it’s hard to add theoretical statements or proofs.

---

### Author Response · Authors · 2023-02-18
**General Response**

We acknowledge the efforts that all reviewers spent on their reviews. Before we respond to the reviewers individually, we first address three shared concerns.

- **Novelty**: The reviewers criticized the paper for lacking novelty (as a video diffusion paper) and requested comparisons with more recent video diffusion models from the past 10 months. However, we still stress that our approach is indeed novel and–as we claimed–concurrent with Ho et al. (2022)’s well-known “Video Diffusion Models” paper from March/April 2022. We asked the **Action Editor to verify this claim** (as can be easily done) since we cannot point to the evidence without revealing our identity. Once the paper is accepted, we will add a discussion on how our approach connects to the more recent literature from the past few months in the camera-ready version.

- **Design of RVD**: The reviewers pointed out that the iterative denoising steps in the diffusion model can be inefficient. We acknowledge this limitation, but argue that linear scaling in the number of diffusion steps T and the video frame length N is unavoidable in autoregressive video diffusion. We stress that model efficiency is less important for video generation than other tasks that require real-time output and provide additional information about model size, training time, and testing time in Section 3.2.

- **Presentation of CRPS**: we acknowledge that our presentation of CRPS could be improved and revise our description to emphasize that CRPS is just one of the three metrics we used. We also note that CRPS is a common metric in classical time-series prediction. We use it to complement FVD and LPIPS to evaluate the performance of our model.

Summary of changes:
- We added a new baseline, SLAMP, from ICCV 2021.
- We added an ablation study, where we compared generating the frame residuals relative to the last frame ($x_{t-1}$) as opposed to the predicted frame ($\mu_t$).
- We added additional discussions about why IVRNN requires longer training sequences than our method.
- We added additional references on video prediction papers.
- We added partial results on SLAMP, which we couldn’t finish due to computational limitations. We will add complete results in the final version.
- ~~**We will upload the revised paper in the next two days.**~~

---

### Decision · Action_Editors · 2023-03-12

**Recommendation:** Reject

**Comment:**

This submission introduces a new way of using denoising diffusion models for video generation.  There are a lot of discussions among the authors, reviewers, and editors.  Eventually, the discussion converges to three major concerns: the novelty wrt. Ho et al, the claim on CRPS, and the missing baselines and references.

The authors communicated with the editors-in-chief and the action editor about the novelty claim. I understand the concern: after another indeed concurrent paper was accepted to a conference (in this case, NeurIPS 2022), the authors are in an awkward position to claim their novelty. In this particular case, I communicated with the reviewers to suggest that it is acceptable to claim Ho et al as a concurrent work; the reviewers agreed and suggested that the lack of comparison with Ho et al. will not be the reason for rejection. However, we also all agree that the authors should cite and discuss Ho et al. (and other papers in the past few months) in detail in the submission.

Regarding the claim on CRPS, the authors have tuned down their claim and I think the revision is acceptable.

Regarding the missing baselines, the authors have included one paper from ICCV 21. However, they have not included other papers listed by the reviewers. Neither have they included another benchmark in their experiments.  The reviewers found such a lack of comparison unacceptable. For example, reviewer 11nc said in the discussion,

"The authors appeared unwilling to cite and include other (very reasonable) baselines during the rebuttal period (e.g, Reviewer eJfF requested DIGAN, reviewer z5Zc requested MCNet, PredNet, and DrNet, I requested HVP by Lee et al.). Instead they included a SLAMP baseline which does not seem as competitive as some of the methods mentioned by the reviewers."

At the end of the discussion, we all feel the paper can be accepted after a revision, if the authors can include additional comparisons and baselines as requested by the reviewers, and systematically discuss related works including Ho et al.  However, as TMLR only allows a minor revision (or rejection), and all reviewers think the change is too major, we have to recommend a "reject and resubmit".

The authors are strongly encouraged to resubmit the paper by including these discussions of related work (it is fine to say they are concurrent if that's indeed the case) and additional comparisons with prior work (e.g., not comparing with Ho et al. is fine, but it should be discussed) and on more datasets, as suggested by the reviewers.

**Audience:**

Yes

**Claims And Evidence:**

The claims are partly supported by the evidence.  Please see the comments below on how the claims can be better supported.